# An estimation of the absolute number of axons indicates that human cortical areas are sparsely connected

**Burke Q. Rosen** [1]*, **Eric Halgren** [1,2]

**1** Neurosciences Graduate Program, University of California San Diego, La Jolla, California, United States of America, **2** Departments of Neurosciences & Radiology, University of California San Diego, La Jolla, California, United States of America

\* bqrosen@health.ucsd.edu

## Abstract

The tracts between cortical areas are conceived as playing a central role in cortical information processing, but their actual numbers have never been determined in humans. Here, we estimate the absolute number of axons linking cortical areas from a whole-cortex diffusion MRI (dMRI) connectome, calibrated using the histologically measured callosal fiber density. Median connectivity is estimated as approximately 6,200 axons between cortical areas within hemisphere and approximately 1,300 axons interhemispherically, with axons connecting functionally related areas surprisingly sparse. For example, we estimate that <5% of the axons in the trunk of the arcuate and superior longitudinal fasciculi connect Wernicke's and Broca's areas. These results suggest that detailed information is transmitted between cortical areas either via linkage of the dense local connections or via rare, extraordinarily privileged long-range connections.

## Introduction

The major tracts connecting cortical areas have long been central to models of information processing in the human brain [1]. Such models have been refined and applied to development and disease with the advent of diffusion MRI (dMRI) [2]. However, dMRI only provides relative connectivity, not the absolute number of axons. Relative connectivity is very useful in many circumstances, but more constraints would be possible if the absolute number of axons connecting different cortical areas could be estimated. Here, we describe and apply a novel method for translating from dMRI-derived streamlines to axon counts.

The ratio between these 2 measures is obtained by comparing the number of streamlines pass through the corpus callosum to the number of axons, as measured with histology. The corpus callosum is uniquely well suited for this purpose. Electron microscopy (EM) can be used to unambiguously count the total number of callosal axons because the limits of the callosum are well defined, axons are aligned, and sections can be cut perpendicular to their axis. Likewise, dMRI-derived streamlines can be unambiguously and exhaustively assigned to the callosum because the source and destination of all axons passing through the callosum as a

**Data Availability Statement:** The processed data used in this study are available at https://doi.org/10.5281/zenodo.6097026. Datafiles are in Matlab v7.3 format. The Human Connectome Project's raw

imaging data and FreeSurfer outputs may be downloaded from https://db.humanconnectome.org.

**Funding:** This work was supported by the by National Institute of Mental Health grant RF1MH117155 (EH) and National Institute of Neurological Disorders and Stroke grant R01NS109553 (EH). The funders had no role in the study design, data collection and analysis, decision to publish, or preparation of the manuscript. The funders had no role in study design, data collection and analysis, decision to publish, or preparation of the manuscript.

**Competing interests:** The authors have declared that no competing interests exist.

**Abbreviations:** AF/SLF, arcuate/superior lateral fasciculus; dMRI, diffusion MRI; EM, electron microscopy; HCP, Human Connectome Project; WM, white matter.

whole are well defined. In contrast to the ipsilateral fiber tracts, in which the majority of axons enter or leave the tract at point between the fascicular terminals, commissural axons must all be connecting the 2 hemispheres. Fortunately, despite these differences, the mean and variability of cross-sectional axon density of the corpus callosum are quite similar to that of telencephalic white matter (WM) in general [3–5], permitting the streamline:axon ratio calculated from callosal fiber to be applied to intrahemispheric connections.

It has long been recognized that as the number of cortical neurons increases, maintaining the same probability of connectivity between neurons would require that axon number increase approximately with the square of neuron number, and this would require too much volume, impose an unsustainable metabolic load [6], and actually decrease computational power due to conduction delays [7]. The consequent imperative to minimize long-distance corticocortical fibers has been posited to be reflected in exponential decline in cortical connectivity with distance [8] and to be partially compensated for with a small-world graph architecture [9], granting special properties to rare long-distance fibers in a log-normal neural physiology and anatomy [10]. However, this organizing principle is rarely explicitly addressed in terms of individual axon counts, nor in a manner both granular and exhaustive with respect to cortical areas. In our histologically calibrated dMRI-derived estimation of these intercortical axon counts, we find that the widespread cortical integration implied by behavioral and mental coherence, and routinely observed in widespread physiological synchronization, belies a surprising small absolute number of long-range axons connecting cortical areas.

## Materials and methods

The basic principle of our method is very simple. Given a dMRI-based measure of total interhemispheric connectivity in arbitrary units and the physical number of axons traversing the corpus callosum, the conversion factor between the 2 can be obtained by dividing the first by the second. Specifically, we started with the total interhemispheric tractography strength reported in our dMRI connectome of the Human Connectome Project (HCP) cohort [11]. For each individual, the cross-sectional area of the corpus callosum was obtained using the standard FreeSurfer structural MRI pipeline [12]. Multiplying this number by a histologically ascertained callosal fiber density [3] yields an estimate of the number of axons traversing an individual's corpus callosum. Dividing this count by individuals' interhemispheric dMRI streamline value yields the conversion ratio from the arbitrarily scaled dMRI metric to the absolute number of axons. Note that this procedure is independent from the scale of the dMRI metric, requiring only that it be proportional to the absolute number of fibers. A moderate proportionality has been observed in comparisons of dMRI and retrograde tracing in macaque [13,14]. Therefore, while the ratio itself is study specific, the procedure can be applied to any dMRI tractography method or parameter set, provided that the dMRI method returns a continuous distribution of connectivity values and has reasonably similar sensitivity to callosal and ipsilateral fiber tracts. To demonstrate this, we repeated the analysis using data from an alternate tractography of the HCP cohort [15]. For detailed methodology, see S1 Appendix.

## Results

Our previous dMRI study [11] includes estimated tractography streamlines between all 360 parcels of the HCP-MMP1.0 atlas [16] for each of the 1,065 individuals in the HCP dMRI cohort. The sum connectivity between the 180 left hemisphere cortical parcels and the 180 right ($180^2$ parcel pairs) constitutes the total callosal connectivity, on average $1.25 \times 10^8$ streamlines. Based on our assumed fiber density of $3.7 \times 10^5$ axons/mm$^2$ [3] and measured callosal cross-sectional area (mean = 689.45mm$^2$), we estimate this cohort to have $2.6 \times 10^8$

callosal fibers on average. The mean quotient between these 2 quantities is 2.00 axons per streamline, a ratio specific to the dMRI methodology and parameters used.

Applying the conversion factor to the interparcel connectivity from our prior study yields an estimate of the absolute number of axons connecting different cortical areas (Fig 1A), $2.43 \times 10^9$ axons in total. This implies that <22% of the approximately $11.5 \times 10^9$ cortical pyramidal cells [17,18] project outside their parcel with the remainder being short-range horizontal or U-fibers. Furthermore, because 51% of interareal axons are to adjacent areas, <11% of pyramidal cells project beyond the adjacent parcel. Axons are approximately log normally distributed among nonadjacent parcel pairs, with adjacent pairs having disproportionally high axon counts (Fig 1B). Median connectivity is approximately 6,200 axons between cortical areas in the same hemisphere and approximately 1,300 interhemispherically. The number of axons in

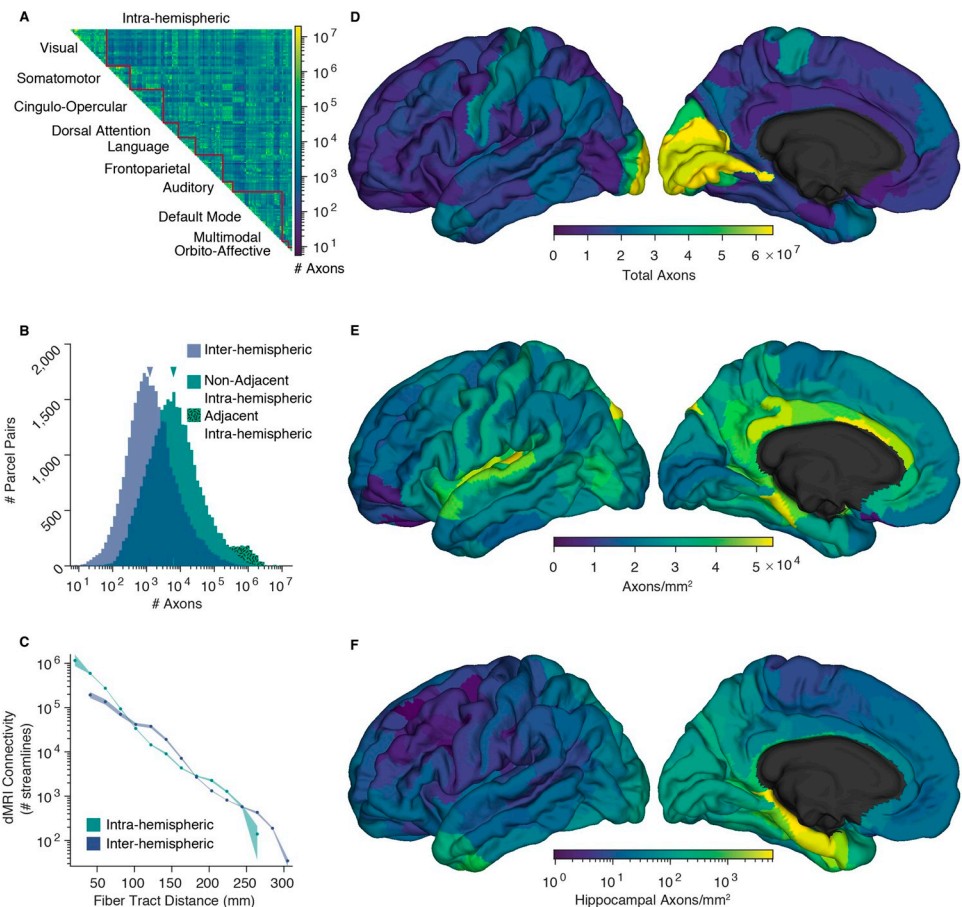

**Fig 1. The number of axons estimated to interconnect the 360 cortical parcels of the HCP-MMP1.0 atlas. (A)** Connectivity matrix of intrahemispheric axon counts, averaged across the 2 hemispheres. Parcels are ordered into 10 functional networks. **(B)** Histograms showing the distribution of inter- and intrahemispheric pairwise axon counts. Physically adjacent and nonadjacent parcel pair intrahemispheric histograms are stacked. Median connectivity, indicated, is approximately 6,200 axons between cortical areas in the same hemisphere and approximately 1,300 interhemispherically. **(C)** Comparison of intra- and interhemispheric dMRI connectivity as a function fiber tract distance. Pairwise values averaged within 15 fiber length bins. Shading shows bootstrapped 95% confidence intervals. **(D–F)** Axon counts and densities averaged across the 2 hemispheres and visualized on the left fsaverage template cortex [12]. (D) Total axons connecting each parcel to all others. (E) Axons connecting each parcel to all others, normalized by the reference parcel's area. (F) Axons connecting the hippocampus to the rest of the cortex, normalized by the area of the cortical parcel, shown in log scale. HCP, Human Connectome Project. The source data for this figure can be found at https://doi.org/10.5281/zenodo.6097026.

the median interhemispheric connection is approximately 20% of that in the median intrahemispheric connection, similar to that found with histological tracing in macaques [19]. The number of callosal axons is not significantly affected by participant sex, although participants ages 26 to 30 have slightly more interhemispheric axons than participants ages 22 to 15 (S2 Fig). This sparse long-range connectivity is consistent with its exponential falloff measured with dMRI in humans and histologically in other mammals [8,11] and with previous statistical estimates based on traditional neuroanatomy [20]. In comparison to the number of axons that would be necessary for complete interconnectivity of neurons in different cortical areas with each other, the estimated number is approximately $10^{11}$ times smaller (see S1 Appendix for calculation).

Our method requires that dMRI tractography performs in a roughly similar manner when applied to intra- versus interhemispheric connections. We evaluated this similarity by comparing linear regressions of distance-matched log-transformed streamlines and found no difference between the slope or intercept of inter- and intrahemispheric connections (Fig 1C); within the common distance domain a paired r test showed no difference in correlation, $p = 0.58$. The total count of a parcel's axons to or from all other areas is much less variable when normalized by the parcel's area (Fig 1D and 1E). Multiplying the total number of fibers by the effective fiber cross-sectional area (the inverse callosal packing density, $2.7 \times 10^{-6}$ mm$^2$/axon) yields $6.6 \times 10^3$ mm$^2$ of cortex or 3.7% of the total white gray interface. Note that the effective axonal area includes myelin, supporting cells and intercellular space in addition to the axon proper. The cross-sectional packing density of human prefrontal WM is quite similar to callosal values with an average of $3.5 \times 10^5$ myelinated axons/mm$^2$ [4] after correction for tissue shrinkage, and this varies among prefrontal regions by a less than factor of 2 [5]. The percent of fiber-allocated cortical area is quite similar to the approximately 4% of total cortical fibers Schüz and Braitenberg estimated [20] are contained in the corpus callosum and long fascicles. The remaining approximately 95% of the cortical gray white interface area is likely occupied by the short range U-fiber system which is difficult to assess with dMRI.

We estimated the fraction of axons traversing the entire length of the arcuate/superior lateral fasciculus (AF/SLF) between termination fields centered on Broca's and Wernicke's areas. The total number of AF/SLF axons was derived by multiplying the tracts' cross-sectional areas by mean axon density of ipsilateral tracts. Using published estimates of the cross-sectional areas of 160.6 mm$^2$ and 51.5 mm$^2$ for the left and right AF and 213.8 mm$^2$ and 174.4 for the left and right SLF [21], and a shrinkage-corrected axon density of $3.5 \times 10^5$ axons/mm$^2$ [4], yields a total of $1.3 \times 10^8$ and $0.8 \times 10^8$ axons in the left and right AF/SLF. These values were compared to the number of tractography-derived axons connecting AF/SLF termination fields [16] according to consensus definitions from reference [22]; see S3 Fig. When the anterior termination field, centered on Broca's area is defined as HCP-MMP1.0 parcels 44, 45, 6r, IFSa, IFSp, and FOP4 and the posterior termination field, centered on Wernicke's area as parcels PSL, RI, STV, and PFcm, trans-terminal axons account for only 0.6% and 0.8% of tract axons in the left and right hemispheres. If the termination fields are liberally expanded to also include parcels 47l and p47r in Broca's and parcels PF, PFm, and PGi in Wernicke's then these percentages increase to a still modest 1.9% and 2.9% of total tract axons. Please note that in this calculation, all axons between Broca's and Wernicke's areas are assumed to pass through the AF/SLF; to the degree that some pass outside the AF/SLF, our estimates should be decreased.

The volume of hemispheric WM occupied by axons between cortical parcels is equal to the sum of all axons' lengths multiplied by their cross-sectional areas. The mean fiber tract lengths of connections were taken from our prior dMRI analysis [11]. For the effective cross-sectional area of axons, we again assumed an effective cross-sectional area equivalent to the inverse of the callosal fiber density [3]. So calculated, the volume occupied by corticocortical and

hippocampocortical fibers is $4.3 \times 10^5$ mm$^3$ or about 96% of the total MRI-assessed WM volume. This implies that the number of long-range fibers cannot be larger than our estimate unless the axon density calculated from the corpus callosum histology is mistakenly low, but this is a linear effect that would need to be unrealistically inaccurate for it to change our main conclusions. Another possibility is that axon density was correctly measured for the corpus callosum but is higher for intrahemispheric fibers and the number of axons per streamline is higher intra- than interhemispherically. These both need to be the case because there is no space available in the hemispheric WM to contain more axons unless they are smaller. However, the histological data indicate that axonal density is approximately the same intra- and interhemispherically [3,4], and the comparison of distance-matched streamlines (Fig 1C) suggests that the number of axons per streamline intra- and interhemispherically is also approximately equal. The remaining WM volume may seem insufficient for connections with subcortical structures. However, the major subcortical structures known to communicate with the cortex (thalamus, amygdala, striatum, nucleus basalis, locus coeruleus, etc.) contain, in sum, less than 1% of the number of cortical neurons [23–25]; see S1 Appendix. Even under the unlikely assumption that all excitatory thalamic neurons project to the cortex with a density across all cortical parcels proportionally to their area, their axons would only comprise approximately 4% of the total WM volume.

We conducted a series of simulations examining the consequences of possible errors in axonal packing density and axon-to-streamline ratio. As noted above, these parameters are coconstrained by the physical volume available for WM. At our derived ratio and assumed packing density, the volume of interareal axons is just under the observed total WM volume. Large net errors in the assumed packing density would result in total interareal axon volumes that are inconsistent with the hemispheric WM volume, a value that is well established. Consequently, in our simulations, axonal packing density and axon-to-streamline ratio were reciprocally changed in order to maintain a constant total cerebral WM volume. Since our parameter estimates are based primarily on fibers passing through the callosum, which are longer in general than intrahemispheric fibers, parameter inaccuracies are more likely for short fibers. In order to evaluate the effect of such inaccuracies, we first simulated the effect of assuming that dMRI sensitivity is systematically reduced (and thus axon count underestimated) for shorter fibers. Uniformly doubling the number of axons with lengths <40 mm in this way yields only a 37.5% increase in the total number of interareal axons and because rank order of observations is not altered the median axon counts are unchanged. Conceivably, our parameter estimates are accurate for the longest fibers (which tend to pass through the callosum) but are progressively less accurate for shorter interareal distances. Consequently, we simulated assuming that our packing density and dMRI sensitivity estimates at the longest fiber lengths were as in our base model and then increased the dMRI streamline-to-axon ratio linearly as the fibers got shorter. Axon density was adjusted to maintain WM volume constant. Even with unrealistically large systematic and matched errors in the 2 estimates, median interareal axon count are only modestly increased; see S4 Fig. For example, adjustments resulting in twice the axon density and half the sensitivity of dMRI to axons at short interparcel distances increase the median number of interparcel axons by only approximately 36%. As reviewed below, errors in interareal axon count due to misestimation of axonal packing density are likely to be relatively unbiased with respect to interareal fiber length, and reported measures vary less than a factor of 2. Our simulations indicate that such errors would have only marginal effects on the median number of interareal axons.

In order to demonstrate that the key principles and findings of this report are robust to the details of tractography procedure, we repeated our analysis on the tractography data of Arnatkeviciute and colleagues [15]. These data consist of a 972-participant subset of the HCP cohort

and use the same parcellation but a different method to reconstruct the fiber streamlines. These data contain fewer streamlines and the estimated numbers of axons per connection are broadly comparable, being somewhat fewer but within an order of magnitude of our primary estimate. We find $8.4 \times 10^8$ total axons and $1.1 \times 10^8$ callosal axons per participant on average, with medians of approximately 1,100 and approximately 130 axons for pairwise intra- and interhemispheric connections between cortical areas (S1 Fig), i.e., even more sparse than those calculated from our tractography data. These data confirm that, independent of tractography procedure, cortical areas are sparsely connected.

## Discussion

In this study, we estimated the absolute number of axons interconnecting cortical areas by calibrating dMRI-based tractography using the histologically ascertained cross-sectional fiber density of the corpus callosum and found that long-range corticocortical connections are quite sparse. Our method depends on histological estimates of callosal axon packing density and leverages the unique properties of the callosum. It assumes proportionality between the number of axons connecting 2 areas and the number of dMRI tractography streamlines for the areal pair, and it assumes approximate parity between dMRI sensitivity to inter- and intrahemispheric connections. In order to estimate the volume occupied by interparcel corticocortical connections, we make the further assumption that the effective axonal packing density is reasonably uniform across the hemispheric WM.

Although there are few reports of axonal density in the corpus callosum [3,26], they are consistent with each other and with reports of axonal density of intrahemispheric tracts [4]. Our study relies on the data reported by Aboitiz and colleagues [3], because it is the most systematic count of which we are aware. There is a later study, [26], which provides a value only slightly lower, $2.83 \times 10^5$ versus $3.7 \times 10^5$ axons/mm$^2$ after correction for tissue shrinkage [27]. However, since this later study was primarily a survey of axon diameter with packing density only incidentally recorded, it is conceivable that density values derived from could be small underestimates. Misestimated tissue shrinkage is possible but likely to be <10% and callosal areas were measured with a well-validated and widely used in vivo method [12]. The axon count estimate does not require that the packing densities of the corpus callosum and ipsilateral WM be the same, but rather the lesser assumption that the axon to dMRI streamline ratio be reasonably uniform across the various instances of long-distance corticocortical connectivity. However, if the packing densities of ipsilateral and callosal long-distance connections are also similar, then the total hemispheric WM volume provides an absolute constraint on the number of long-distance connections, and this provides a powerful validation of our estimate.

The literature provides converging evidence that the fiber packing density of WM varies by at most a factor of 2 across the cortex, including the corpus callosum. A histological study of human cortex found only a 7% difference in the axon densities of the callosum versus the superior longitudinal and inferior occipitofrontal fasciculi [26]. Zikopoulos and Barbas [4] found that the cross-sectional packing density of human prefrontal WM is remarkably similar to callosal values with an average of $3.5 \times 10^5$ myelinated axons/mm$^2$ after correction for tissue shrinkage and that this varies among prefrontal regions by a less than factor of 2 [5]. In addition, while dMRI-based estimates of axon density and caliber are imperfect, they suggest that axon density varies by less than 2-fold both between major ipsilateral tracts and within each tract along their length [28]. While packing density is not directly equivalent to axon caliber, the 2 are likely inversely related. Axon diameter, as estimated with dMRI, varies by 20% among ipsilateral tracts and by at most a factor of 2.2 among WM voxels, including those of the corpus callosum [29,30]. Overall, distributions of dMRI-derived axon diameter for the

callosum [31] and whole cerebrum [30] are similar with the bulk of values between 2.5 and 5 μm. Histological measurements in macaques concur that axon diameters are very similar within the callosal and noncallosal segments of major fasciculi and vary by less than 2-fold across the cortex [32]. Furthermore, even this limited variation in diameter is not systematically dependent on the length of axons but rather on their regions of origin and termination [33].

Concerning dMRI to axon count proportionality, it has been shown that there is a moderate linear correlation between dMRI-traced streamlines and the number of fibers identified with histological tracers connecting cortical regions in macaques [13,14] and a strong correlation in ferrets [34]. Based on these data, Donahue and colleagues [13] concluded that dMRI tractography was capable of quantitively describing corticocortical WM tracts with approximately order of magnitude precision using a high-quality dataset such as the HCP. In line with previous statistical estimates of whole-cortex axon counts based on traditional neuroanatomy [20], the numbers of axons in pairwise connections are probably correct within an order of magnitude. Estimates of this precision are useful as we find that that interareal connectivity derived from dMRI varies over more than 7 orders of magnitude.

While we derive a single dMRI to axon count factor, it is likely that the true conversion ratio varies somewhat among connections due to local microstructural differences other than axon count such as axon caliber, packing density, or myelination. However, the major fasciculi (including the corpus callosum) have similar axon calibers and packing densities, varying by only a factor of approximately 2 across the cortex when examined in humans histologically [4,5,26] or with dMRI [28–30] and histologically in macaques [32]. More generally, packing density is a function of local cellular interactions, especially with oligodendrocytes, and because these are the same in the corpus callosum and intrahemispheric WM, the a priori expectation is that packing density would be similar, as histological data support. Myelination has only a modulatory effect on dMRI-detected anisotropy, with most of the effect derived from axonal membranes [35]. Nevertheless, microstructural variation in dMRI to axon count ratio may be a source of noise in our estimates. Our simulations of misestimation of dMRI to axon ratio and packing density show that these errors only marginally affect estimated axon counts and do not alter our conclusions. Our findings were also similar when we repeated our analysis on an alternative, somewhat sparser dMRI tractography dataset [15], demonstrating that the details of the tractography algorithm do not affect the overall tenor of our results. While the scope of this study is limited to long-range fibers, we note that shorter, more superficial U-fibers are systematically less myelinated and of lesser caliber than their interareal counterparts [5], and, therefore, the procedure outlined here may require modification before being applied to them.

As previously stated, this methodology assumes a reasonable degree of parity in the sensitivity of the dMRI tractography to intra- and interhemispheric fiber tracts. Consistent with this assumption, we found little difference between distance-matched dMRI connectivity for callosal versus ipsilateral connections. If callosal axons were more easily detectable, this would be reflected as an upward displacement of the interhemispheric trace (blue) above the intrahemispheric trace (green) across the entire distance domain in Fig 1C, which is not evident. This parity is perhaps unsurprising because over most of their trajectories, interhemispheric fibers are subjected to the same crossing fiber issues as intrahemispheric. Specifically, the corpus callosum is a distinct tract for only about 15 to 35 mm, but its fibers range in length up to about 300 mm. Thus, the fraction of an interhemispheric tract that resides within the callosum is inversely proportional to its total length. Consequently, if there were enhanced detection of callosal fibers as streamlines by dMRI, one would expect the interhemispheric (blue) trace in Fig 1C to be elevated primarily at short fiber lengths and depressed at long fiber lengths,

resulting in a noticeably steeper slope for the interhemispheric than the intrahemispheric trace, which is not observed. These data support the applicability of the scaling factor derived from interhemispheric fibers to intrahemispheric fibers.

The corpus callosum was used to calibrate the estimate because of its unique properties: It has a well-defined cross-sectional area, more than approximately 99% of interhemispheric corticocortical axons are routed through it [3,36], and essentially no fibers leave or enter the tract between the 2 hemispheres. This is in sharp contrast to noncommissural fasciculi. While it may be commonly assumed that within major cortical fasciculi the majority of axons terminate or originate at the ends of the tract and thus carry information along its entire length, an alternative conception is that these large bundles are composed mostly of axons shorter than the total fascicular length which enter and exit the tract at various points. By analogy, the former assumption likens a tract to a tunnel, where all traffic is trans-terminal, whereas the latter conceives of tracts as like interstate highways, where very few vehicles travel the entire route. In a supplementary analysis, we compared these models for the AF/SLF system. The total number of AF/SLF axons was estimated using the tract diameters [21] and packing densities [4] taken from the literature. The number of trans-terminal axons was determined by defining termination field parcels centered on restricted and inclusive definitions of Broca's and Wernicke's areas [22]. Depending on the assumptions, only about 1% to 5% of the axons in a middle section of these fasciculi are trans-terminal. These percentages may be overestimates since they assume that all fibers between the posterior and anterior areas travel through the AF/SLF. However, even if these values are a 2-fold underestimate, it suggests that only a small fraction of the axons in the ipsilateral fasciculi are trans-terminal. The evidence indicates that the "highway" model is more apt for the ipsilateral fiber tracts, and this conception is consistent with neural wiring being driven, in large part, by exponential distance rules [8]. This of course, does not apply to the corpus callosum, as there is no interterminal cortex to project into.

Importantly, the reconceptualization of major intrahemispheric tracts as containing few fibers connecting their distant terminals is still consistent with the long and well-established impression from blunt dissection [37] and dMRI orientation maps [38] that the hemispheric WM is largely composed of well-defined long-distance tracts. Indeed, we estimate that approximately 96% of the WM is composed of interareal axons. What these observations suggest is that the major tracts arise from the tendency of axons to grow in alignment with axons that are already present using established mechanisms of adhesion and fasciculation [39]. Thus, axons are free to join tracts at various points, and tend to proceed together (fasciculate), but again are free to leave whenever they approach their own target. In other words, axons are joined in a given tract because they share a direction rather than an origin and destination.

The interareal axon counts we derive here permit other interesting quantitative estimates that may inform models of cortical neurophysiology. For example, the connections between Wernicke's and Broca's areas are thought to integrate receptive and expressive aspects of language, but we estimate that there are only approximately 58,000 axons between the core cortical parcels in these regions (parcels 44 and PSL), fewer than 2 for each word in an average university student's vocabulary [40]. Another example where quantitative appreciation of direct axonal connections may influence neurocognitive models are the hippocampocortical interactions subserving recent memory, which are commonly posited to carry information regarding the contents of the memory trace during memory formation, consolidation, and retrieval. We estimated that areas distant from the hippocampus, notably the dorsolateral prefrontal cortex, may be connected to it by $<10$ axons/mm$^2$ (Fig 1F), including both efferent and afferent axons, yet hippocampo–prefrontal interactions are considered crucial for contextual recall [41]. We estimated the average neural density in the cortex as approximately 92,300 neurons/mm$^2$ by dividing the $16.34 \times 10^9$ cortical neurons (including interneurons) from [17] by

the $1.77 \times 10^5$ mm$^2$ mean white gray surface area of the HCP cohort used. If the sparse connectivity suggested by our calculations is correct, it implies that hippocampo–dorsolateral prefrontal interactions in memory are likely mediated by polysynaptic pathways.

These constraints encourage consideration of models of cortical function where connections are dense but mainly local, i.e., a small-world network with intense interconnections within modules and sparse projections between them. While this general principle is widely accepted, the scale of the vast gulf in absolute connectivity between local and long-range connections is startling. This network architecture provides for the wide and efficient distribution of information created by local processing within modules [8,9]. A more uniformly connected cortex would require more WM, necessitating a more voluminous cerebrum and the human cortex is near the limit after which an increase in size reduces computational power [6,7]. Functionally, a deep reservoir of weak connections enables a large number of states and eases state transitions [10]. Modeling suggests that long-range covariance and even synchrony can be achieved through activation of multisynaptic pathways rather than direct connections [42,43], and possible signs of these have been observed experimentally in humans [44,45].

While the long-range direct corticocortical axons are few in number, we note that axons are heterogeneous and that these counts are a limited proxy for true interareal connectivity. Axons, especially those connecting architectonically similar regions, may have a disproportionate impact on the flow of information despite their rarity [46–48], by virtue of their morphology (e.g., greater diameter, larger termination fields, greater axonal arborization, or more numerous en passant varicosities) or by molecular synaptic specializations. For example, while only approximately 5% of the synapses to V1 layer 4 come from the lateral geniculate body in the macaque [49], they have an outsized effect on their firing [50]. The importance of rare intermodule connections might also be enhanced if they are focused on a small location within cortical parcels (i.e., the rich club [51]), but this has not been convincingly demonstrated. Last, it is useful to note that these quantitative considerations are radically different in other species, where the smaller number of cortical neurons and shorter interareal distances allow greater connectivity between cortical areas, as well as a larger proportion of subcortical connectivity [7].

## Supporting information

**S1 Appendix. Extended methods and results.**
(PDF)

**S1 Fig. Replication results.** Results obtained by repeating the analysis using an alternative tractography dataset [15]. **(A)** Connectivity matrix of intrahemispheric axon counts, averaged across the 2 hemispheres. Parcels are ordered into 10 functional networks [11]. **(B)** Histograms showing the distribution of inter- and intrahemispheric pairwise axon counts. Physically adjacent and nonadjacent parcel pair intrahemispheric histograms are stacked. Median connectivity, shown in gray, is approximately 2,500 axons between cortical areas in the same hemisphere and approximately 300 interhemispherically. Parcel pairs with zero axons connecting them are represented by the bars left of the y-axis. **(C)** dMRI connectivity as a function fiber tract distance. Pairwise values averaged within 15 fiber length bins. Shading shows bootstrapped 95% confidence intervals. **(D–F)** Axons counts and densities averaged across the 2 hemispheres and visualized on the left fsaverage template cortex [12]. (D) Total axons connecting each parcel to all others. (E) Axons connecting each parcel to all others, normalized by the reference parcel's area. (F) Axons connecting the hippocampus to the rest of the cortex, normalized by the area of the cortical parcel, shown in log scale. dMRI, diffusion MRI.
(PDF)

**S2 Fig. Effects of sex and age on the interhemispheric connectivity.** Each individual's total number of estimated interhemispheric axons is shown with a marker. The black horizontal bars show the group means and vertical bars the bootstrapped 95% confidence intervals these means. Shading shows a kernel density estimate of the group distributions. Note that while the number of pairwise axons is approximately log normally distributed across areal pairs, it is approximately normally distributed across individuals. The only significant group difference is between the 22 to 15 and 25 to 30 age groups, $F_{1,1064} = 7.646$, $p = 0.0058$. Interactions between sex and age effect were not significant. As we assume a constant fiber density, our estimate the total number of interhemispheric fibers is a linear multiple of the callosal cross-sectional area. (PDF)

**S3 Fig. AF/SLF termination fields.** In order to estimate the fraction of tract axons that travel the entire length of the tract, its termination fields, centered on Broca's and Wernicke's areas, were manually defined in terms of HCP-MMP1.0 parcels [16] according to consensus definitions [22]. **(A)** Conservative definitions where the anterior termination field, in teal, is composed of parcels 44, 45, 6r, IFSa, IFSp, and FOP4 and the posterior termination field, in yellow, is composed of parcels PSL, RI, STV, and PFcm. For this definition, trans-terminal axons account for 0.6% and 0.8% of tract axons in the left and right hemispheres. **(B)** Liberal definitions in which parcels 47l and p47r were added anteriorly and parcels PF, PFm, and PGi were added posteriorly, resulting in trans-terminal axons accounting for 1.9% and 2.9% of tract axons in the left and right hemispheres. Gray lines indicate the approximate locations at which the tract diameter, used to estimate the total number of tract axons were ascertained [21]. Areas are rendered on the fsaverage template cortex [12]. AF/SLF, arcuate/superior lateral fasciculus; HCP, Human Connectome Project. (PDF)

**S4 Fig. Simulated effect of short axon underestimation on pairwise axon counts.** In order to explore the possibility that dMRI tractography is less sensitive to shorter connections the streamline-to-axon ratio was increased linearly with inverse fiber length using a range of slope parameters. Packing density was assumed to be reciprocally decreased in order to fix a constant total cerebral WM volume. **(A)** Distributions of intra- and interhemispheric interareal axons counts. Gray histograms show the effect of uniformly halfling the ratio and doubling the assumed packing density. **(B)** Intrahemispheric and **(C)** interhemispheric adjustments to the # of pairwise axons as a function of inverse fiber tract length and the resultant increase in median axon count as a function of the adjustments' slope parameter. At slope = 0, values are unadjusted from the primary analysis. Large adjustments only increase median counts modestly in the context of the log-normal distribution. For example, for intrahemispheric connections (B), at a slope of 0.004, the number of the shortest axons is about doubled (i.e., corresponding to doubling the axon density and halving the sensitivity of dMRI to axons), but the median number of interparcel axons only increases by approximately 36%. dMRI, diffusion MRI; WM, white matter. (PDF)

## Acknowledgments

Data were provided, in part, by the Human Connectome Project, WU-Minn Consortium (Principal Investigators: David Van Essen and Kamil Ugurbil; 1U54MH091657) funded by the 16 National Institutes of Health (NIH) Institutes and Centers that support the NIH Blueprint for Neuroscience Research and by the McDonnell Center for Systems Neuroscience at Washington University.

## Author Contributions

**Conceptualization:** Eric Halgren.

**Data curation:** Burke Q. Rosen.

**Formal analysis:** Burke Q. Rosen.

**Funding acquisition:** Eric Halgren.

**Investigation:** Burke Q. Rosen.

**Methodology:** Burke Q. Rosen.

**Software:** Burke Q. Rosen.

**Supervision:** Eric Halgren.

**Visualization:** Burke Q. Rosen.

**Writing – original draft:** Burke Q. Rosen, Eric Halgren.

**Writing – review & editing:** Burke Q. Rosen, Eric Halgren.

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
