## [Editor Report · Decision Letter 0]

12 Aug 2021

Dear Dr Rosen, 

Thank you for submitting your manuscript entitled "Human cortical areas are sparsely connected: Combining histology with diffusion MRI to estimate the absolute number of axons" for consideration as a Short Report by PLOS Biology.

Your manuscript has now been evaluated by the PLOS Biology editorial staff, as well as by an academic editor with relevant expertise, and I am writing to let you know that we would like to send your submission out for external peer review.

Please re-submit your manuscript within two working days, i.e. by Aug 16 2021 11:59PM.

Kind regards,

Gabriel Gasque

Senior Editor

PLOS Biology

ggasque@plos.org

---

## [Decision Letter · Decision Letter 1]

16 Sep 2021

Dear Dr Rosen,

Thank you very much for submitting your manuscript "Human cortical areas are sparsely connected: Combining histology with diffusion MRI to estimate the absolute number of axons" for consideration as a Short Report at PLOS Biology. Your manuscript has been evaluated by the PLOS Biology editors, by an Academic Editor with relevant expertise, and by three independent reviewers. Reviewer 2 is Basilis Zikopoulos and reviewer 3 is Almut Schüz.

The reviews of your manuscript are appended below. You will see that the reviewers find the work potentially interesting. However, based on their specific comments and following discussion with the Academic Editor, I regret that we cannot accept the current version of the manuscript for publication. We remain interested in your study and would be willing to consider resubmission of a comprehensively revised version that thoroughly addresses all the reviewers' comments. We cannot make any decision about publication until we have seen the revised manuscript and your response to the reviewers' comments. Your revised manuscript would be sent for further evaluation by the reviewers.

Reviewers 2 and 3 expressed significant enthusiasm about the approach, while reviewer 1 raises several methodological concerns. We think that the request made by reviewers 1 and 2 that the relationships established for the fibers of the human corpus callosum need to be carefully cross validated for other fiber systems, and ideally in animal models where ground truth tractography data are available, would be strictly required for a successful revision. Together with the Academic Editor, we think this point is critical, but may be difficult to address, as it depends on the availability of matched histological and noninvasive diffusion imaging data. However, we do not think that such a cross validation can be postponed to a later study, as one of the reviewers suggested. First, as the reviewers pointed out, projection systems are indeed highly diverse in terms of axonal diameters, myelination, trajectories, etc., and what is true for callosal projections does not necessarily need to hold for short- or long-range corticocortical projections within the hemispheres. Second, if the present paper is published, we expect it to be widely cited and used in many further analyses and modeling; therefore, the numbers presented here need to be reliable. Without these additional analyses, we will not be prepared to move forward with your manuscript.

You will see that in addition there is a long list of specific methodological issues put forward by the three reviewers, but we think you should be able to address most of them through additional analyses or with explicit caveats in an extended discussion. This includes the appropriate scaling of the conversion factor –which is central to this study– that was debated by reviewer 3. As this reviewer states, a scaling of the conversion factor by factor two may not actually alter the principal conclusions of the paper; nonetheless, proper accounting for tissue shrinking of the used literature data is still important.

We appreciate that our requests represent a great deal of extra work and are willing to relax our standard revision time to allow you six months to revise your manuscript.

**IMPORTANT - SUBMITTING YOUR REVISION**

*Resubmission Checklist*

*Published Peer Review*

*PLOS Data Policy*

*Blot and Gel Data Policy*

Sincerely,

Gabriel Gasque

Senior Editor

PLOS Biology

ggasque@plos.org

REVIEWS:

Reviewer #1: The authors present a potentially interesting paper that proposes a relationship between probabilistic tractography streamlines and absolute axon counts after having related the data to the prior histological literature, particularly Aboitiz et al., 1992, but also Arcelli et al., 1997 and Liewald et al., 2014. I am not convinced of the authors' claims at this point. 

Major Concerns:

1) The authors cite Donahue et al., 2016 Journal of Neuroscience as evidence of a strong relationship between histological and tractography connection strengths. However, their data shows a strong correlation only for the short distance strong connections (see their Figs 4A and 5). Long-distance (e.g., such as the corpus callosum) and weak connections had many discrepancies between tractography and tracers (see Figure 4a). Thus, I am not sure that this citation is good evidence for the authors' point. 

2) The authors use a single value for the total number of axons crossing the corpus callosum; however, it is known (e.g., shown in Aboitiz et al., 1992) that there are major differences in axonal density in different parts of the corpus callosum, with larger, lower density axons in the posterior midbody and posterior splenium of the corpus callosum with higher axonal density in the genu and rostrum. If we are to believe that tractography streamlines can be scaled to represent axonal density, we would expect a similar pattern to be present in the tractography data as the axons cross the midsagittal plane of the corpus callosum. Was this indeed the case in the study carried out by the authors? Also, it is not clear to me that the authors have internalized this important point from Aboitiz et al., given the statement "However, the major fasciculi (including the corpus callosum) have similar axon calibers and packing densities," as it would seem that the corpus callosum at least is not a good example of such properties. 

3) It was not immediately clear to me how the authors derived axonal values from Arcelli et al., 1997 and Liewald et al., 2014 on a quick review of these papers. 

4) I had some concerns about the tractography methods: a) It seems that the authors did not use the surface-based tractography available in probtrackx2 despite the fact that it is more accurate than the 3D volume-based approach (e.g., voxel corners do not stick out into deep white matter) and the 360 cortical areas exist natively on cortical surface meshes. b) More to the point, the authors seem not to have understood how to use the data as released by the HCP for such purposes and produced a convoluted and potentially error prone process for getting the 360 cortical areas aligned with the bedpostX data that requires mapping from the 32k fs_LR space to fsaverage ico5 space, then to the FreeSurfer volume space, and then finally to the bedpostX space. Even if they had chosen not to use surface-based tractography, they could simply have mapped the cortical areas directly from the 32k fs_LR surfaces in the ${StudyFolder}/${Subject}/T1w/fsaverage_LR32k folder to the volume space in the ${StudyFolder}/${Subject}/T1w/Diffusion.bedpostX folder using a single command wb_command -label-to-volume-mapping. A simple question on the HCP-Users mailing list could have clarified the correct approach before large scale computations had been undertaken. c) Similarly, it was not clear how the authors prevented axons from crossing CSF spaces bounded by the pial surface or through the ventricles. Thus, it is not clear that the streamlines being studied here were forced to even cross the corpus callosum. 

5) Inadequate consideration of alternative explanations. The authors state key assumptions regarding fiber diameter distributions and the 'conversion ratio' in several places: 

a. line 81 "Note that this procedure does not at all depend on the scale of the dMRI metric, requiring only that it be proportional to absolute number of fibers." 

b. line 84: "It only requires that the dMRI method return a continuous distribution of connectivity values and have reasonably similar sensitivity to callosal and ipsilateral fiber tracts." 

c. Line 153 "While we derive a single dMRI-to-axon count factor, it is possible that the true conversion ratio varies somewhat among connections due to local microstructural differences other than axon count such as axon caliber, packing density, or myelination. However, the major fasciculi (including the corpus callosum) have similar axon calibers and packing densities [12]."

The main experimental support of these statements is the Liewald et al. (2014) EM study reporting roughly similar fiber diameter distributions for two large intrahemispheric tracts compared to the callosum. However, an alternative hypothesis is that short-distance pathways, particularly those connecting sulcal banks with a thin white matter gyral blade in between are mediated by axons having substantially smaller average diameter. Suppose, for example, that there is a two-fold difference in average diameter for axons shorter than ~4 cm (never reaching the major tracts) vs axons that do reach major tracts. Since these account for the majority of intrahemispheric connections (Fig. 1C), this would imply a many-fold greater number of inter-areal axons than the 1.05 x 10^9 value proposed by the authors. In principle, the disparity between short and long distance axonal diameters could be even greater. Hence, without high quality empirical data on axonal diameters as a function of axonal length, it behooves the authors to be far more circumspect in their claims.

6) Issues of scaling. There is an apparent mismatch between the reported tractography-based streamline distances and the physical size of the human brain.

a. Line 112: "The mean fiber tract lengths of connections were taken from our prior dMRI analysis [3]." This mean length is not stated in the present ms and does not appear to have been mentioned in ref 3. Please state what that value is. 

b. Line 145 "Specifically, the corpus callosum is a distinct tract for only about 15-35 mm, but its fibers range in length up to about 300 mm". This is consistent with Fig 1C of the current study, which shows a maximum length >300mm for interhemispheric and >250mm for intrahemispheric fiber tract distance. This is very puzzling insofar as the maximum tract length shown in Fig. 7 of Rosen & Halgren 2020 is <180 mm, which is consistent with the known A-P length of human brains. The authors need to explain the apparent discrepancy between the two studies and to provide evidence that these extreme lengths represent plausible anatomical trajectories within the white matter.

7) Confusing wording. Line 145 "Of course, short interhemispheric trajectories are dominated by the callosal segment, and long by the intrahemispheric segments. Generally, the proportion of their trajectories that are within the callosum is roughly proportional to their length. Consequently, if there were enhanced detection of callosal fibers as streamlines by dMRI, the green trace in figure 1C would be expected to be elevated primarily at short fiber lengths, resulting in a noticeably steeper slope for the blue than the green trace, which again is not observed. These data support the applicability of the scaling factor derived from interhemispheric fibers to intrahemispheric fibers."

These statements are confusing - please explain them more clearly.

8) Tractography false positives. Fig 1D: V1 has one of the highest apparent interhemispheric connectivity values among all parcels. Yet in the macaque few if any V1 neurons project contralaterally (Van Essen et al., J. Neuroscience, 1982). Hence, this is a likely example of artifactually large number of false positive connections revealed by tractography.

Reviewer # 2, Basilis Zikopoulos: The paper entitled "Human cortical areas are sparsely connected: Combining histology with diffusion MRI to estimate the absolute number of axons" provides estimates of the number of long-range axons connecting cortical areas within and between hemispheres, using the myelinated fiber density of the corpus callosum, measured histologically, to calibrate whole-cortex diffusion MRI (dMRI) connectivity data. The authors conclude that cortical areas within hemispheres are sparsely connected, with an average of about 2,700 axons, whereas connections of areas between hemispheres involve about five times fewer axons. 

This is a very timely, interesting, straight-forward, and well written study and I want to commend the authors on their excellent idea to perform this, much needed work, and analysis. The strengths of this manuscript include (1) the high quality of the data analyzed (from detailed high-resolution histological studies and a large dMRI dataset) and (2) the theoretical framework used to correlate data from sets at multiple scales for the analysis, which is based on the key relationship between structural features and connectivity of the cortex. 

The main apparent limitations that will need to be addressed in this or future experimental and theoretical work include the lack of cross-validation of the findings using additional approaches for calibration and combination of the two datasets, the dearth of high-quality, high-resolution histological data, and the low resolution and threshold sensitivity, especially of the dMRI dataset for the detection of short- and some medium-range connections that also include significant numbers of thin, branching, and unmyelinated axons. Despite these limitations, the findings are novel, highly significant, and the manuscript is poised to be an outstanding contribution to our field and of general interest. 

Below few comments and suggestions that in my opinion will further increase the value and clarity of the manuscript, strengthen reported findings, and place them in the context of key principles that underlie functional and structural cortical network organization and connectivity: 

1. The last sentence in the abstract (lines 59-61) and relevant discussion part (last paragraph, lines 180-188) are somewhat problematic, or a narrow interpretation that can be misinterpreted or misleading and should be modified. The authors' interpretation that the sparseness of long-range connectivity suggests that cortical integration relies mainly on extremely dense local connections and that models that require direct long-range connectivity are somehow challenged by these findings, is not justified, because that would assume that all pathway interactions with different types of excitatory and inhibitory neurons, receptors, distal or proximal dendritic segments, numbers of axon branches and terminals are similar across the cortex. Another equally plausible interpretation based on these findings would be that sparse, in terms of axons, long-range connectivity can still produce major effects postsynaptically, which can integrate information and direct cortical activity, due to specialized interactions with key elements of local circuits. In addition, the last statement of the abstract also suggests that direct transmission of information between cortical areas may be substituted by indirect connections (serial multisynaptic steps of short-range connections) overlooking differences in conduction velocity and variable key interactions with distinct local inhibitory and other microenvironments for each set of pathways. As such, the following statement in the discussion "In such models, cortical locations interact through activation of multi-synaptic pathways rather than direct connections, and the key to connectivity is physiological selection under multiple constraints rather than anatomical projections" wrongly implies that long-range pathways do not lead to connectivity that is physiologically selected under multiple constraints and complex interactions, and should be modified. 

2. Methods and Results sections: the authors compare linear regressions of distance-matched streamlines to find no difference between inter- and intrahemispheric connections, as shown in Fig. 1C. Even though this is a previously used approach, it must be noted that although distance is often correlated with the strength and/or presence of connections in some studies, it is well-established that it doesn't accurately and fully capture the relationship between connections, and in many cases, it falls apart when describing long-range connections, especially between frontal and parietal lobes, which together constitute a large chunk of the cortex. On the other hand, structural (dis)similarity between areas is a much better predictor of connectivity strength in most mammals studied, including primates, as shown in several studies using golden-standard tract-tracing and structural imaging approaches (see relevant work by Hilgetag, Barbas, Zikopoulos etc.). As the authors likely know, the Structural Model for Connections, also known as Architectonic Type Principle, was initially proposed by Helen Barbas after multiple tract tracing studies in non-human primates; later, this model was extended to other mammalian species and, since then, predictions based on the Structural Model have been consistently confirmed across all cortical lobes and systems in non-human primates and other species. Therefore, we can assume that the relational principle of the Structural Model obtained from animal research applies to the human cerebral cortex. Based on this large body of work we know that cortical areas tend to be connected primarily with other areas that are relatively similar in type (structure etc.), most of which happen to be nearby, but some are quite distant, as for example in the case of the relatively strong connections between lateral prefrontal and parietal cortices that do not fit and overrun distance models. Therefore, several tract tracing studies in non-human primates show that long-range cortico-cortical connections across lobes are far from weak and do involve lots of axons. See for instance the summary figures (Figures 15 & 16) in Cavada & Goldman-Rakic, 1989 J Comp Neurol 287: 422-445. Cavada and Goldman-Rakic showed that projections from prefrontal areas to posterior parietal areas, which are long range connections, are denser than short-range projections from other areas that are closer to the parietal areas injected by these authors. Actually, dense long-range cortico-cortical connections have been shown for multiple areas across the cerebral cortex of the macaque (see Morecraft et al 2004 J Comp Neurol for posterior cingulate and posterior parietal areas, Morecraft et al 2012 Brain Res Bull for frontal motor and anterior cingulate areas, Morecraft et al 2015 Brain Res Bull for insular and parietal somatosensory areas; Zikopoulos et al 2018 PLOS Biol for prefrontal connections with all other lobes; Cavada et al 2000 Cerebral Cortex for orbital areas; Medalla and Barbas, 2006 for frontal and parietal areas; or Joyce and Barbas, 2018 for especially strong long-range connectivity between anterior cingulate areas in the frontal lobe and area prostriata in the occipital lobe of primates). Based on connectivity patterns in primates, the laminar architecture of the cortex, and the principles of the Structural Model, Zikopoulos et al. in 2018 (PLOS Biol) showed that eulaminate areas have comparatively more and stronger long-range connections than limbic cortices and went one step further to predict that this would also apply to the human cortex, especially since the human cortex includes more eulaminate cortices. These findings from tract tracing studies should be taken into consideration and since this study deals primarily with long-range connectivity it would be appropriate and more accurate to correlate dMRI connectivity with structural features of the connected parcels, if possible. Several recent studies in humans have parcellated the human cortex using relevant structural features that could be used to explore dMRI connectivity relationships. If not possible, at the very least, this should be briefly noted in the Discussion or Appendix. 

3. The axon density in the corpus callosum estimated in previous histological studies (Aboitiz et al., or Liewald et al.) was used and was combined with dMRI data to estimate axon numbers in pathways. The second study referenced has also reported axon density in other major long-distance pathways of the brain, including the superior longitudinal fasciculus or the uncinate fasciculus. In addition, several other high-resolution histopathology studies at the light and electron microscopic level have examined and reported axon features and density in the white matter below prefrontal and temporal cortices (e.g. Zikopoulos and colleagues in 2010, or 2018; Liu and Schumann, 2014) that participate in short- or long-range cortical connections. Combined, some of these data on other major white matter pathways in the human brain could be used to calibrate dMRI connectivity data and cross-validate estimates derived from the callosal calibration. Some of these studies also include very relevant information on the relative prevalence of short- vs long-range connections that could support and strengthen the authors' findings. 

4. Discussion, page 7, lines 174-178: this statement can be misconstrued and should be reworded. I recommend stating that <5% of cortical pyramidal cells project outside their immediate neighborhood instead of using the word "area". This is because many short-range white matter connections that are below the resolution of dMRI approaches and are not included in this analysis are between adjacent, relatively small areas or neighboring columns within an area. In addition, a study by Zikopoulos et al., 2018 in PLOS Biology has also shown a clear relationship between the number of neurons and the density of white matter pathways in non-human primates and humans and could be relevant to this statement. 

5. The authors show that interhemispheric connectivity constitutes on average about 20% of intrahemispheric connectivity (540 vs 2,700 axons). This estimate is in line with similar estimates from Barbas et al., 2005 for non-human primates (estimated that less than 30% of connections are contralateral in rhesus monkeys). 

6. Since this study focuses on analysis of long-range connectivity, and uses long-range callosal connections to calibrate dMRI data, it is important to highlight key, relevant differences between long- and short-range connections and U-fibers, something that is now missing. The authors do briefly state that dMRI methods cannot reliably resolve short-range connections, but I think the readers would appreciate a specific comment in the discussion/supplement regarding key features of pathways, other than axon density and number such as: proportion of thin vs thick axons that correlate well with short- and long-range pathways, proportion of myelinated vs unmyelinated axons (more unmyelinated axons in limbic cortical pathways), myelin thickness data for short- vs long-range connections, which is relevant for conduction velocity and function, axonal branching patterns and size of termination fields that may disproportionately amplify the effects of some connections over others, and how some of these features change as we move from the superficial to the deep white matter (e.g. see Liewald et al., 2014; Zikopoulos and Barbas 2010; Zikopoulos et al., 2018; Caminiti and Innocenti, 2009; LaMantia and Rakic 1990; Makris et al., 1999; Rademacher et al., 1992).

7. Reference list (page 10): references 5 and 11 refer to the same study and one of them should be deleted, and in-text citations corrected appropriately. Same for references 12 and 14. 

8. S1 Appendix, page 14, lines 359-360: The authors state that they used estimates on thalamocortical connections from histological counts in reference [12] - Arcelli P, Frassoni C, Regondi MC, De Biasi S, Spreafico R. GABAergic neurons in mammalian thalamus: a marker of thalamic complexity? Brain Res Bull. 1997;42: 27-37. As I have carefully read this paper several times I am not sure how the authors got estimates about volume of thalamocortical axons and total axon count (22.6x106) from this study, which only reports local inhibition in the thalamus. Perhaps they meant to cite reference [11] instead - Ji JL, Spronk M, Kulkarni K, Repovš G, Anticevic A, Cole MW. Mapping the human brain's cortical subcortical functional network organization. Neuroimage. 2019;185: 35-57. doi:10.1016/j.neuroimage.2018.10.006. Please clarify and correct, as needed. 

Reviewer #3: 

Review by Almut Schüz, see attached file

Summary

This is a fascinating paper. It quantifies in an elegant way cortico-cortical connections between distant cortical areas in the human brain. The results are in support of findings which indicate a preponderance of connectivity between closely located areas (Schüz and Braiten- berg, 2002), also in other species (Scannell et al., 1995; Schüz at al., 2006). The study by Rosen and Halgren is outstanding since - in contrast to previous studies – it is able to provide an astonishingly concrete estimate for the median number of axons between distant cortical areas in the human brain.

The study uses the parcellation into 180 areas in each hemisphere by Glasser et al. (2016). This parcellation is based on a combination of neuroanatomical (mainly myelin) and functional features, by way of MRI and fMRI. The number of areas comes close to that of the myeloarchitectonic areas by the Vogt and Vogt school. The present study is based on diffusion MRI data from the database of the Human Connectome Project.

In this paper, the relative connectivity provided by dMRI (number of streamlines) is transformed into absolute numbers of axons. This transformation is based on a comparison with histological data from the literature on the density of axons in the Corpus callosum (Aboitiz et al, 1992). It leads to a conversion factor of 0.87 axons per streamline. The authors assume that the same factor can be applied to both, the Corpus callosum and to the other long range systems via the white matter. This is a reasonable assumption.

Presentation of data:

It would be good to visualize not only the median:

The HCP data contain a family structure with genetic related and unrelatedness and many

other behavioral measures (Van Essen et al., 2012). The data also vary with age (22-36 years). Thus, the number of streamlines (Page 11, Line 292) between cortical areas shows inter- individual variability, affecting axon estimation. It would be essential to visualize a scatter

plot (e.g. for the Corpus callosum) how the spread of the number of axons is depicted in the healthy HCP sample. Are these values in an acceptable range? As tractography relies on coarser spatial resolution, partial volume effects, and may be erroneous due to false- positive/negative estimation of streamlines.

The conversion factor

The conversion factor is the crucial point in this paper. Re-reading Aboitiz’ paper and based on my own histological experience I come to the conclusion that your cobersion factor is at the lower end and is rather around 1.6. This does not invalidate the paper – a factor of 2 is negligible in this kind of statistical neuroanatomy – but it gives an idea of the possible range.

Let me explain. In Line 307 to 313 you describe your approach. In line 308 you say “electron microscopic study”, but it is both light and electron microscopic. The shrinkage factor mentioned in the method’s part of Aboitiz’ paper is only valid for his light microscopic material, embedded in paraffin. He does not mention any shrinkage factor for his electron microscopic material (embedded in Epon), and – according to our own experience – there is hardly any shrinkage in such material. (The volume in our EM-material is about 96% of the original tissue after fixation; Schüz and Palm. 1989).

The number you mention for light microscopy of 1.57x105/mm2 is not mentioned explicitly in

Aboitiz paper as far as I can see, but you probably calculated it from the data given in his table I and corrected it for areal shrinkage. Correct?

Aboitiz estimates that about 20% of fibers were not detected in the light microscope. So we end up with a range of about 1.6x105/mm2 from light microscopy and about 3.8 x 105/mm2 from electron microscopy. The reality is probably somewhere between these values.

This is supported when looking at the total number of axons in the Corpus callosum. Aboitiz estimates 2 x108 fibers. This is twice the number you get when using his light microscopic density of about 1.6x105mm2 and your average areal size. (He does not give an areal size as

far as I can see). This speaks in favour of a density between the LM and EM-data, and it leads to a conversion factor of 1.6 rather than 0.87.

The inverse packing density (area per axon) in line 104 would then be lower, but well within the possible range. The average axonal diameter is below 1 m in most cortico-cortical long- range systems (Liewald et al, 2014).

Some points to be clarified

In the discussion in lines 130 and in line 165 the authors quote Liewald et al. (2014) for an alternative value for packing density in the corpus callosum of 1.23 x 105 /mm2. I cannot find this number in the quoted paper. Did the authors somehow calculate this value from the fiber diameters given there? Or did I overlook something?

Another point: in line 176 the authors quote Azevedo et al. (2009) for a number of 11.5x109 cortical pyramidal cells. I cannot find a number for cortical pyramidal cells in this paper. Did the authors derive this from the total number of cortical neurons mentioned on p.535 (16.34x109 ) and perhaps subtract a percentage of non-pyramidal cells?

Also, in some cases the same paper is quoted under 2 different numbers in the reference list: Liewald et al. under 12 and 14, Aboitiz et al under 5 and 11.

Finally, on line 107 the names Schüz and Braitenberg are misprinted. (And thanks to this quotation I discovered a serious printing error in our own paper: on p.381, first line, it should be 6x109 not 6x103)

References:

Aboitiz at al. (1992), as quoted under [5] and [11]

Glasser et al. (2016), as quoted under [7]

Liewald et al. (2014), as quoted under [12] and [14]

Scannell MP, Blakemore C, Young MP (1995) Analysis of connectivity in the cat cerebral cortex. The J. of Neuroscience 15, 1463-1483

Schüz A., Chaimow D, Liewald D and Dortenmann M (2006) Quantitative Aspects of

Corticocortical Connections: A Tracer Study in the Mouse. Cerebral Cortex October 2006;

16:1474--1486 , doi:10.1093/cercor/bhj085

Schüz and Braitenberg (2002), as quoted under [8]

Schüz A. and Palm G (1989) Density of neurons and synapses in the cerebral cortex of the mouse. The J. Comp. Neurol. 286: 442-455

Glasser, M.F., Coalson, T.S., Robinson, E.C., Hacker, C.D., Harwell, J., Yacoub, E., Ugurbil, K., Andersson, J., Beckmann, C.F., Jenkinson, M., Smith, S.M., Essen, D.C.V., 2016. A

multi-modal parcellation of human cerebral cortex. Nature 536, 171–178. https://doi.org/10.1038/nature18933

Van Essen, D.C., Ugurbil, K., Auerbach, E., Barch, D., Behrens, T.E.J., Bucholz, R., Chang, A., Chen, L., Corbetta, M., Curtiss, S.W., Della Penna, S., Feinberg, D., Glasser, M.F., Harel, N., Heath, A.C., Larson-Prior, L., Marcus, D., Michalareas, G., Moeller, S., Oostenveld, R., Petersen, S.E., Prior, F., Schlaggar, B.L., Smith, S.M., Snyder, A.Z., Xu, J., Yacoub, E., WU- Minn HCP Consortium, 2012. The Human Connectome Project: a data acquisition perspective. NeuroImage 62, 2222–2231. https://doi.org/10.1016/j.neuroimage.2012.02.018

---

## [Editor Report · Decision Letter 2]

8 Feb 2022

Dear Dr Rosen,

Thank you for submitting your revised Short Report entitled "Human cortical areas are sparsely connected: Combining histology with diffusion MRI to estimate the absolute number of axons" for publication in PLOS Biology. I have now discussed this new version with the Academic Editor. I am pleased to let you know that we will probably accept this manuscript for publication, provided you satisfactorily address the remaining minor points raised by the Academic Editor, which are included below my signature. Please also make sure to address the following data and other policy-related requests:

1) Title: 

We would like to suggest a more direct title that might be more appealing to a broad readership. We recommend: “An estimation of the absolute number of axons indicates that human cortical areas are sparsely connected”

However, we would be happy to work with you on an alternative if you think our suggestion misrepresents your findings.

2) Data:

2.1) Please upload to Zenodo the data for all supporting figures and include a README file that explains how these data were analyzed to generate the plots and graphs displayed in those figures.

2.2) Please also ensure that each figure legend in your manuscript includes information on where the underlying data can be found: https://doi.org/10.5281/zenodo.5204805

3) Peer review history: 

We think it would be extremely useful for future readers to make the peer-review history of your manuscript accessible, so they can read and assess your thoughtful answers to the reviewers. You will be offered this option later on during the production process, and we ask that you take it.

We expect to receive your revised manuscript within two weeks. 

*Published Peer Review History*

*Early Version*

Sincerely,

Gabriel

Gabriel Gasque, Ph.D.,

Senior Editor,

ggasque@plos.org,

PLOS Biology

Academic Editor's comments:

To reiterate the main point of this paper, the authors compared post-mortem axon density in the human corpus callosum with estimates of callosal streamline density from non-invasive diffusion tractography and found that their calculations suggest that most cortical areas in the human brain are linked by a surprisingly small absolute number of projection neurons. As the authors state, this is a result that will be very widely debated and referenced and may form the basis of a reevaluation of models of cortical activity and function, which currently assume a more massive neuronal communication system. (Metaphorically speaking, right now we assume that the cortex is quite well-connected by a far-reaching multi-lane highway system, when in fact there may exist just a few foodpaths between most areas. Naturally, this change of perspective strongly affects how we understand signalling mechanisms in the brain.) The finding crucially hinges on the inferred conversion factor of streamlines to axons. Therefore, the reviewers and authors were chiefly concerned with getting this conversion factor right. I think the authors have made a reasonable case that, while there are several constraints on the reliability of the conversion, the conversion factor that they derived is largely accurate.

Generally, I think that the authors responded to the reviewers’ comments very thoroughly, and while they did not take up every suggestion, they addressed the essence of the criticism that centered on the correct determination of the conversion factor between DTI streamlines and axons. I do think the authors have done a good job in this respect, reassuring the reviewers that their calculations are at least correct within an order of magnitude. There are still a number of caveats (particularly concerning whether the relations established for the corpus callosum translate also to other fiber systems within each cortical hemisphere), but at least these caveats are explicitly addressed in the manuscript. Moreover, there is a main issue of interpretation of the findings which was brought up by R2, of whether the findings imply that cortical signals only propagate between adjacent areas, or whether long-distance projections may have a functional role after all. These issues will likely be the subject of many studies starting from these findings, and they are more cautiously discussed now.

As a minor point, the authors may find this additional reference useful, which I think further strengthens their case: Delettre et al. 2019, Comparison between diffusion MRI tractography and histological tract-tracing of cortico-cortical structural connectivity in the ferret brain. Netw Neurosci 3(4):1038-1050. doi: 10.1162/netn_a_00098.

Moreover, I was confused by this statement of the authors in their response letter: “…the data in figure 5A indicates that, relative to histological tracing, dMRI tends to underestimate the presence of long-range connections. This implies that long range cortical connectivity may be even sparser than our dMRI-based evidence indicates…” — I think they mean “overestimate”.

---

## [Editor Report · Decision Letter 3]

17 Feb 2022

Dear Dr Rosen,

On behalf of my colleagues and the Academic Editor, Claus Hilgetag, I am pleased to say that we can in principle accept your Short Report "An estimation of the absolute number of axons indicates that human cortical areas are sparsely connected" for publication in PLOS Biology, provided you address any remaining formatting and reporting issues. These will be detailed in an email that will follow this letter and that you will usually receive within 2-3 business days, during which time no action is required from you. Please note that we will not be able to formally accept your manuscript and schedule it for publication until you have any requested changes.

PRESS

Sincerely, 

Gabriel Gasque, Ph.D. 

Senior Editor 

PLOS Biology

ggasque@plos.org